# Bayesian hierarchical modeling of sea level extremes in the Finnish coastal region

Olle Räty[1], Marko Laine[1], Ulpu Leijala[1], Jani Särkkä[1], and Milla M. Johansson[1]

[1]Finnish Meteorological Institute, P.O. BOX 503 FI-00101 Helsinki, Finland

**Correspondence:** Olle Räty (olle.raty@fmi.fi)

**Abstract.** Occurrence probabilities of extreme sea levels required in coastal planning, e.g. for calculating design floods, have been traditionally estimated individually at each tide gauge location. However, these estimates include uncertainties, as sea level observations typically have only a small number of extreme cases such as annual maxima. Moreover, exact information on sea level extremes between the tide gauge locations and incorporation of dependencies between the adjacent stations is often
lacking in the analysis.

In this study, we use Bayesian hierarchical modeling to estimate return levels of annual maxima of short-term sea level variations related to storm surges in the Finnish coastal region. We use the generalised extreme value (GEV) distribution as the basis and compare three hierarchical model structures of different complexity against tide gauge specific fits. The hierarchical model structures allow to share information on annual maximum sea levels between the neighboring stations and also provide
a natural way to estimate uncertainties in the theoretical estimates.

The results show that compared to the tide gauge specific fits, the hierarchical models, which pool information across the tide gauges, provide narrower uncertainty ranges for both the posterior parameter estimates and the corresponding return levels in most locations. The estimated shape parameter of the GEV model is systematically negative for the hierarchical models, which indicates a Weibull-type of behavior for the extremes along the Finnish coast. The negative shape parameter also allows us to
calculate the theoretical upper limit for the annual maximum sea levels on the Finnish coast. Depending on the tide gauge and hierarchical model considered, the median value of the theoretical upper limit was 47–73 cm higher than the highest observed sea level.

## 1 Introduction

Extreme sea level phenomena (waves, storm surges, tides, etc.) together with the rising mean sea level introduce hazards
both to people and coastal infrastructure by causing migration, loss of functionality and biodiversity, and by decreasing our living habitat. In the Baltic Sea region, sea level extremes are directly associated with such hazards as coastal erosion and flooding (e.g., Rutgersson et al., 2022; Weisse et al., 2021). Recent studies have shown that the increase in the mean sea level has generally exceeded the global average during the past 50 years in the Baltic Sea (Weisse et al., 2021) with some local exceptions from this trend (e.g., Männikus et al., 2020). It is foreseen that the main drivers for long-term changes in the extreme
sea levels are changes in the the relative mean sea level and atmospheric conditions. For these reasons, reliable projections of

extreme sea levels are key tools for supporting coastal planning and safety in regional scales. For example, estimation of low-probability - high-consequence events such as extremely high sea levels are important for nuclear power plant safety in Finland (Jylhä et al., 2018).

The Finnish coast is surrounded by the Baltic Sea – an intra-continental small sea that is connected to the Atlantic Ocean via narrow and shallow Danish Straits (Leppäranta and Myrberg, 2009). The unique geography of the Baltic Sea as well as various local and global processes govern the sea level variations on the Finnish coast. On short temporal scales, wind and air pressure are the main factors inducing local sea level fluctuations along the Finnish coast. In addition, wind-generated waves, seiche oscillations (standing waves inside the Baltic Sea basin), and meteotsunamis (e.g., Pellikka et al., 2020) (meteorologically induced long, shallow-water waves) alter the sea level locally on a short time scale. Tides (e.g., Medvedev et al., 2013; Särkkä et al., 2017) and ice have a marginal influence on the sea level variations on the Finnish coast compared to other forcing factors. In longer-term perspective, the in- and outflow of the water from the Danish Straits controls the total water balance in the Baltic Sea. Other important long-term elements are the mean sea level change reflecting the behavior of global sea level rise, as well as post-glacial land uplift, which originates from the pressure release of the crust after the last glacial era. The largest sea level variations on the Finnish coastline take place in the ends of the Bay of Bothnia and Gulf of Finland due to the piling up effect and standing wave oscillations within the bay (Jönsson et al., 2008).

Previous studies have addressed coastal flooding risks in the present-day and future climatic conditions at the Finnish tide gauge locations. Recently, Pellikka et al. (2018) estimated future coastal flooding risks by combining mean sea level projections with the short-term sea level variability. Leijala et al. (2018) examined the total water level on the shore due to simultaneous effect of sea level and wind waves. Physical limits of simulated extreme sea levels were investigated in hydrodynamic modeling exercises by Särkkä et al. (2017) on the Helsinki coastal area using winds from regional climate model simulations and by Averkiev and Klevannyy (2010) for the whole Gulf of Finland with idealised wind fields as forcing. Wolski et al. (2014) and Wolski and Wiśniewski (2020) showed that there are large geographical variations in the observed sea level extremes in the Baltic Sea and that they tend to increase towards the ends of the bays in the northern parts of the Baltic Sea domain (see Fig. 1). Johansson et al. (2001) inspected temporal variations in the extreme sea levels on the Finnish coast and concluded that a general increase has occurred in the annual maxima during the 20[th] century. Rapid sea level oscillations due to air-pressure disturbance i.e. meteorological tsunami waves have also been studied on the Finnish coast by concentrating on their occurrence in the Gulf of Finland (Pellikka et al., 2020).

The theory on extreme value analysis is well documented in the literature (e.g., Coles, 2001). If block-maxima, such as the annual maximum sea levels are considered, extreme value theory tells us that the generalised extreme value distribution (GEV) is the only possible limiting distribution. GEV has been used to model return levels of annual maximum sea levels in the Baltic Sea region in many previous studies (e.g., Ribeiro et al., 2014; Wolski et al., 2014; Marcos and Woodworth, 2017; Soomere et al., 2018; Kudryavtseva et al., 2021). Most of these studies, however, have concentrated on individual tide gauge locations. As the time series of sea level extremes are relatively short, they generally do not allow reliable estimation of occurrence probabilities of very rare events (e.g. return period >1000 years). Furthermore, to be able to construct predictions for sea level extremes outside the tide gauge locations, information on their spatial dependencies and variations is needed. Soomere et al.

(2018) inspected spatial variations of GEV parameters along the Estonian coast using ocean model data as the basis for extreme value analysis.

Issues with the limited sample size can be partly alleviated by pooling information on sea level extremes across the neighbouring tide gauges, which reduces uncertainty on the parameter estimates. One approach to pool information across multiple tide gauges is based on regional frequency analysis (RFA) (Dalrymple, 1960; Hosking and Wallis, 1997). In this approach, a reasonably homogeneous region is searched using some similarity criterion, tide gauges within the region are normalised locally using flood-index and then pooled together before fitting a single extreme value distribution to the pooled data. This approach has been successfully applied to storm surges (e.g., Bardet et al., 2011; Bernardara et al., 2011). However, the results obtained with RFA might contain inconsistencies between different domains. Also, the method does not directly account for spatial dependencies between the tide gauges within the whole target region.

Bayesian hierarchical modeling approach allows more flexibly to incorporate spatial and other dependencies in statistical models (e.g., Gelman et al., 2013). In our case, this means that tide gauge specific GEV parameters are described jointly at the population level, for example, assuming that they come from the same joint hyper-distribution. This hyper-distribution has its own hyper-parameters which need to be specified either from data or modeled by adding an additional layer to the model. Bayesian hierarchical models have been used to estimate extremes in other geoscientific fields (e.g., Cooley et al., 2007), but there are relatively few examples of their use when modeling sea level extremes. Coles and Tawn (2005) made one of the first attempts to model storm surges within the Bayesian framework. Calafat and Marcos (2020) developed a novel hierarchical model to account for spatial dependencies on storm surge extremes over the North Sea and British Isles. They showed that Bayesian hierarchical modeling approach allows to estimate occurrence probabilities of sea level extremes outside gauge locations and to quantify uncertainty in the predicted sea levels.

This paper applies previous efforts on statistical modeling of sea level extremes on the Finnish coastal region. Our aim is to assess how Bayesian hierarchical modeling – implemented in a simpler manner compared to Calafat and Marcos (2020) – performs in comparison to tide gauge specific models, when estimating theoretical return levels for extremes related to short-term sea level variations. Three hierarchical models are included, two of which take the spatial distance between the tide gauges into account in their model formulations. We perform a series of tests with the hierarchical models and evaluate their performance against the observed sea level extremes. Furthermore, we illustrate the theoretical return level estimates obtained with the hierarchical models and their differences with respect to tide gauge specific estimates. We exclude long-term changes from the analysis and focus on short-term sea level variations controlled by meteorological factors. As we concentrate solely on stationary conditions, the potential time dependence of GEV parameters is not assessed.

The paper is structured in the following manner. In Section 2, we describe the tide gauge observations from the Finnish coast. In Section 3, we introduce the methods utilised in this paper, which is then followed by a summary of the main outcomes of the study in Sect. 4. In Section 5, we discuss the shortcomings of this study and potential avenues for future research. The paper is closed with conclusions in Sect. 6.

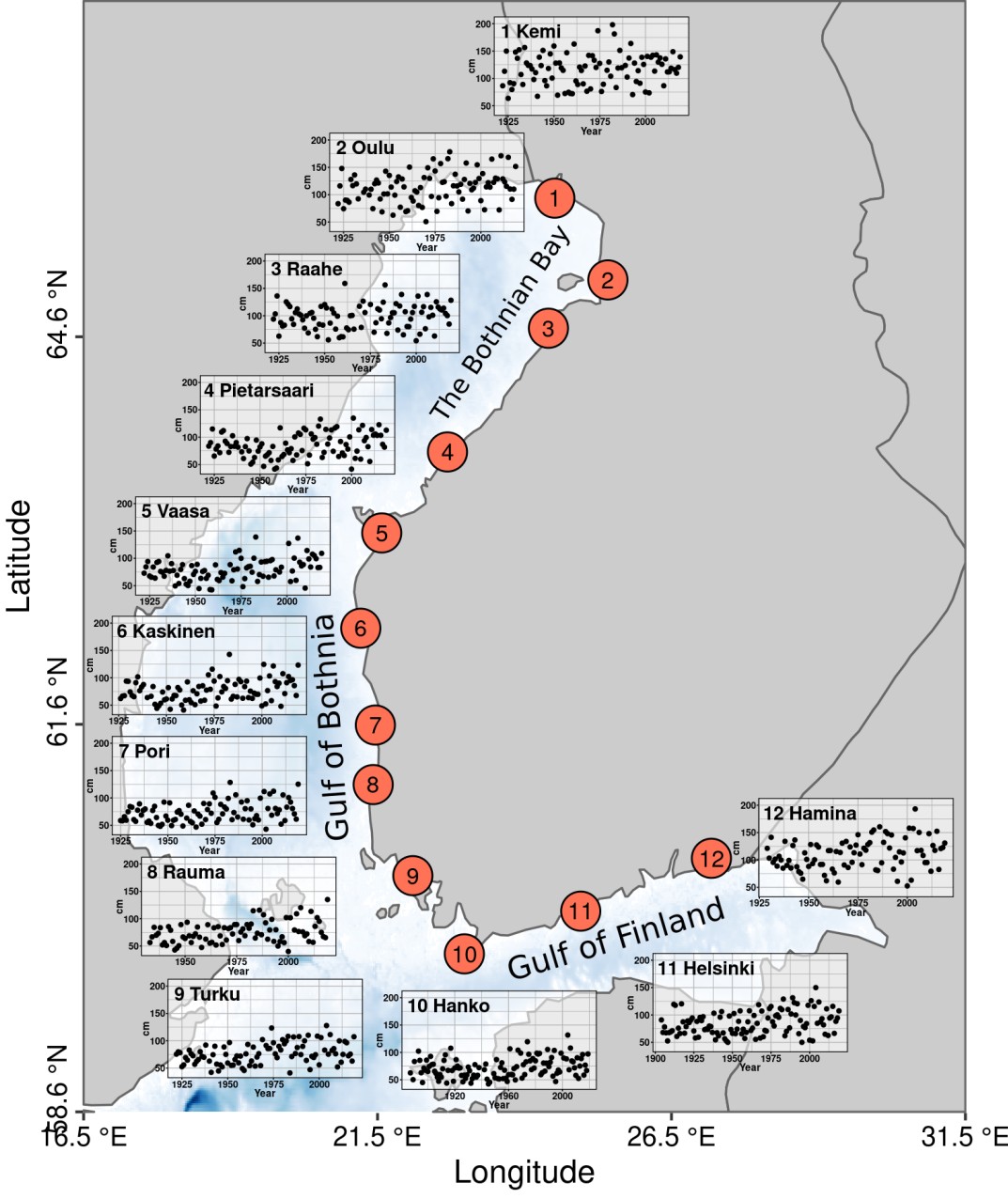

**Figure 1.** Detrended time series of the annual maximum sea level at each tide gauge location selected for this study. The bathymetric data used in the background map is taken from ETOPO1 database (Amante and Eakins, 2009) using the `marmap` package in R (Pante and Simon-Bouhet, 2013).

## 2 Tide gauge records on the Finnish coast

Finnish Meteorological Institute (FMI) maintains and collects sea level observations from 14 tide gauges along the Finnish coast. Most of the Finnish tide gauges have been constructed in the 1920s, meaning that they have more than 90 years of sea level records currently available. The available time series are shorter only for Rauma tide gauge (built in 1933) and for Porvoo tide gauge (built in 2014). The longest time series providing over 100 years of data are available from Hanko (the first tide gauge in Finland, built in 1887) and Helsinki (tide gauge built in 1904). Two tide gauges were excluded from further analysis

either due to the too short length of time series for extreme value analysis (Porvoo) or the differing behavior of the annual maximum sea levels in the tide gauge location in the Finnish archipelago (Föglö) compared to the coastal tide gauges, which was not captured by our simple hierarchical modeling approach.

Data set used here consists of annual maximum sea levels. Annual maxima were calculated from monthly maximum sea levels which were extracted from either the original continuous paper recordings, or later from the digital data at 1-minute

resolution. Before year 1939, the values have been calculated from 4-hourly observations. Although there have been changes in the temporal resolution of the observations, we decided to include the earliest years in our study in order to use the longest possible time series in our analysis. The measured sea levels have been converted to the height system N2000 from fixed tide-gauge-specific reference systems. N2000 is the Finnish realisation of the European Vertical reference system, and the datum has been derived from Normaal Amsterdams Peil (NAP) (Saaranen et al., 2009). All time series extend to the end of year

2020 with some gaps in them due to missing data. To reduce the impact of long-term changes caused by climate change and post-glacial land uplift to the extreme value analysis, a linear trend was calculated separately for each tide gauge based on 4-hourly observations and then subtracted from the annual maximum values.

Studies in the Baltic Sea region have suggested that the annual maximum sea levels should be calculated for a year-long block that covers all winter months instead of using the calendar year due to seasonality in storminess (e.g., Johansson et al.,

2001; Suursaar et al., 2002; Soomere et al., 2018; Kudryavtseva et al., 2021). We took this approach and calculated the annual maximum values between 1 April–31 March. This way the maximum value from each winter period was selected, which removes the correlation between the annual maximum values. Furthermore, those years, which had more than one monthly maximum value missing from the winter half of the year were filtered out. The length of the pre-processed time series varied between 87–124 years. These are illustrated in Fig. 1.

The maximum sea level varies between 36–198 cm in the pre-processed time series. From Fig. 1, it is observed that the highest sea levels are typically observed in Kemi and Hamina, which are the tide gauges located closest to the ends of the Gulf of Bothnia and the Gulf of Finland. Time series for these tide gauges also exhibit the highest year-to-year variability. These features have some implications for the statistical model design as discussed in the next section. We note that the highest annual maximum values used here differ to certain extent from the reported record heights in Finland (e.g., Wolski and Wiśniewski,

2020) due to detrending and as they have been previously defined against a different reference, the so-called theoretical mean sea level (Johansson et al., 2003).

Some individual values stand out from the time series panels. For example, markedly high sea levels were observed in January 1984 on some tide gauges located on the western coast of Finland. Such exceptional cases are of interest to us, because they are likely to some extent affect our statistical modeling results. We will briefly discuss the sensitivity of our GEV models for anomalously high observed sea levels in Sect. 4 using the aforementioned case as an example.

## 3   Methods

### 3.1   Extreme value analysis for annual maximum sea levels

We briefly summarise the main properties of the generalised extreme value distribution (GEV) applied to the extreme sea levels before discussing the chosen modeling approaches. We refer to Coles (2001) for more details on GEV distribution and its statistical properties. Let $Y_i$ with $i = 1, \ldots, 12$, be a random variable describing the annual maximum sea level (i.e., block-maxima) at the $i^{\text{th}}$ tide gauge. The extreme value theorem states that for the normalised maxima of a sequence of independent and identically distributed random variables the GEV distribution is the only possible limiting distribution. GEV is a suitable model in our case, as the detrended annual maxima can be considered to be independent of each other. The cumulative distributions function of GEV can be written as:

$$G(y; \mu, \sigma, \xi) = \exp\left( -\left[ 1 + \xi \left( \frac{y - \mu}{\sigma} \right) \right]_+^{-1/\xi} \right). \tag{1}$$

In Eq. (1), $\mu \in \mathbb{R}$ is the location parameter, $\sigma > 0$ scale parameter and $\xi \in \mathbb{R}$ denotes the shape parameter. The tail behavior of GEV distribution is strongly controlled by the shape parameter. If $\xi < 0$, $y$ has an upper limit at $\mu - \xi/\sigma$, while in other cases the upper tail is unbounded and has either exponential ($\xi = 0$) or polynomial ($\xi > 0$) decay.

The quantiles of GEV distribution are obtained by inverting Eq. (1):

$$y_p = \begin{cases} \mu - \frac{\sigma}{\xi} \left[ 1 - \{ -\log(1 - p) \}^{-\xi} \right], & \text{for } \xi \neq 0, \\ \mu - \sigma \log\{ 1 - \log(1 - p) \}, & \text{for } \xi = 0. \end{cases} \tag{2}$$

This equation is used to calculate the return level $y_p$ associated with a certain exceedance probability $p$ or equivalently, the return period $T = 1/p$. While other approaches such as peaks-over-threshold could be considered (e.g., Coles, 2001), we used GEV as it allows the direct modeling of annual maxima.

In the simplest case where the GEV parameters are estimated separately for each tide gauge (denoted as Separate hereafter), the model for the observations at the $i^{\text{th}}$ tide gauge is expressed as $y_i \sim GEV(\mu_i, \sigma_i, \xi_i)$. We use Bayesian methods to estimate the distribution parameters. As we are particularly interested in robustly quantifying uncertainty in the GEV parameters and in the predicted return levels, the Bayesian approach provides a natural way to do this. Bayes' theorem states that the posterior distribution of model parameters given observations $y$ is

$$p(\theta | y) = \frac{p(\theta) p(y | \theta)}{\int p(\theta) p(y | \theta) \, d\theta} \tag{3}$$

Here, $p(\theta)$ is the prior distribution for the parameters and $p(y|\theta)$ the likelihood function. The symbol $\theta$ denotes a vector that contains all the unknown parameters of the model. As the integral in the denominator in Eq. (3) does not generally have a closed-form solution for our hierarchical models, we use Markov chain Monte Carlo (MCMC) methods to sample from the posterior parameter distribution $p(\theta|y)$. The implementation details such as the selected prior distributions for the parameters $p(\theta)$ (and for the hyper-parameters in hierarchical modeling cases) are provided in the supplementary material.

## 3.2 Hierarchical models for GEV parameters

Tide gauge specific parameter estimates tend to be uncertain, as we usually have relatively few observations in our disposal. One way to include more information is to assume that the model parameters at different tide gauge locations are similar. A hierarchical approach is to assume that their values are bind together with a joint prior distribution, whose parameters are not assumed to be known, but estimated along the individual gauge specific parameters. For pooling information across the tide gauges, we tested three hierarchical formulations for the GEV distribution parameters. The first model (denoted as Common hereafter) is a simple extension to the separate fitting of GEV distribution at the tide gauge locations. The model is written as

$$
\begin{aligned}
y_i &\sim GEV(\mu_i, \sigma_i, \xi_i), \ i = 1, \ldots, 12 \\
\mu_i &\sim \mathcal{N}(\mu_\mu, \sigma_\mu^2) \\
\sigma_i &\sim \mathcal{N}_+(\mu_\sigma, \sigma_\sigma^2) \\
\xi_i &\sim \mathcal{N}(\mu_\xi, \sigma_\xi^2)
\end{aligned}
\tag{4}
$$

where $\mathcal{N}_+$ denotes the half-normal distribution. This model applies partial pooling to the data with the assumption that the tide gauge specific distribution parameters come from the same joint Gaussian distribution but it simultaneously allows different parameter values to be estimated for individual tide gauges. In addition to the tide gauge specific GEV parameters, we have six additional hyper-parameters ($\mu_\mu$, $\sigma_\mu$, $\mu_\sigma$, $\sigma_\sigma$, $\mu_\xi$, $\sigma_\xi$), that tie the individual parameters together. As the hyper-parameters are unknown and estimated from the data, we need to assign further prior distributions on them (see the supplementary material for more details).

## 3.3 GEV parameter hierarchy using splines

Common does not account for possible spatial dependencies in the sea level extremes between the tide gauges. As was mentioned previously, the northern part of the Baltic Sea consists of sub-basins like the Gulf of Bothnia and the Gulf of Finland. This geometry strongly regulates the sea level variations in these regions (so-called bay effect), and the magnitude of sea level extremes systematically increases towards the end of the two bays (Fig. 1). We used the distance $d$ calculated roughly along the coast between the tide gauge locations such that $d_{\mathrm{Kemi}} = 0$ as covariate information when modeling the GEV parameters. In these models, the vector for the GEV parameters is expressed as $\theta = (\mu(d_i), \sigma(d_i), \xi_i)$. Thus, we assume the shape parameter $\xi$ does not depend on $d$ but still gets its own value at each tide gauge, similarly as in Sect. 3.2. One benefit of this approach is that the model provides estimates of the GEV parameters and consequently return level estimates between the tide gauge locations.

We tested two approaches to incorporate the distance dependence in our hierarchical models. In the first one, the location and scale parameter are modeled using B-splines (de Boor, 1978) (denoted as Spline hereafter). B-splines are defined by the degree $p$ of the basis function polynomials and a non-decreasing, here equally-spaced, set of knots $\mathbf{t} = t_1, \ldots, t_r$. We have set $r = 10$. Spline functions are then constructed as a linear combination of B-spline basis functions. The model formulation for Spline is

$$y_i \quad \sim \quad GEV(\sum_{j=1}^m \alpha_j B_j(d_i), \sum_{j=1}^m \beta_j B_j(d_i), \xi_i), \tag{5}$$

where $m = r + 2$ is the number of cubic ($p = 3$) B-splines, $B_j$ are the B-spline basis functions and $\alpha_j$ and $\beta_j$ are the spline coefficients to be estimated from the data. To avoid overfitting, we used cubic B-splines with first order random walk priors for the spline coefficients (e.g., Eilers and Marx, 1996; Lang and Brezger, 2004).

## 3.4  GEV parameter hierarchy using Gaussian processes

In the third hierarchical model, spatial dependence for the GEV parameters is accounted for using a Gaussian process (GP) prior. The model is written as:

$$\begin{aligned} y_i &\sim GEV(f_\mu(d_i), f_\sigma(d_i), \xi_i) \\ f_\mu &\sim \mathcal{GP}(m_\mu, K_\mu) \\ f_\sigma &\sim \mathcal{GP}(m_\sigma, K_\sigma). \end{aligned} \tag{6}$$

In Eq. 6, $m_\mu$ and $m_\sigma$ are the mean and $K_\mu$ and $K_\sigma$ the covariance functions for $\mu$ and $\sigma$, respectively. As we have a finite number of tide gauge locations, this amounts to modelling both priors as multidimensional Gaussian distributions. To obtain smoothness in the neighbouring tide gauge estimates for the GEV parameters, we used squared exponential covariance function in our model:

$$K(d_i, d_j | \alpha, \rho) = \alpha^2 \exp\left[ -\frac{1}{2} \frac{(d_i - d_j)^2}{\rho^2} \right], \tag{7}$$

where $d_i$ and $d_j$ are tide gauge locations defined by their distances to the reference station at Kemi. Furthermore, $\alpha$ is spatial standard deviation and $\rho$ is the characteristic length scale, both of which are estimated from the data.

All models were fitted using MCMC simulations by R and Stan probabilistic programming language (Gabry and Češnovar, 2021; Stan Development Team, 2020a, b). Stan implements a variant of Hamiltonian Monte Carlo MCMC algorithm, so-called No-U-Turn Sampler (NUTS), which has shown to perform well in fitting complex hierarchical models (e.g., Calafat and Marcos, 2020). For each model, MCMC simulations were done with four parallel chains over 3000 iterations. The first 1000 iterations were removed as the burn-in period. Thus, the total number of draws obtained from the posterior distribution was 8000.

Figure 2 shows an example how the Spline and GP fits for the location and scale parameter vary along the tide gauge locations from Kemi to Hamina. Overall, both methods provide smooth, yet flexible fits between the stations. Also the uncertainty estimates given by these methods are very similar. The similarity of the obtained fits is not surprising, as the formulation

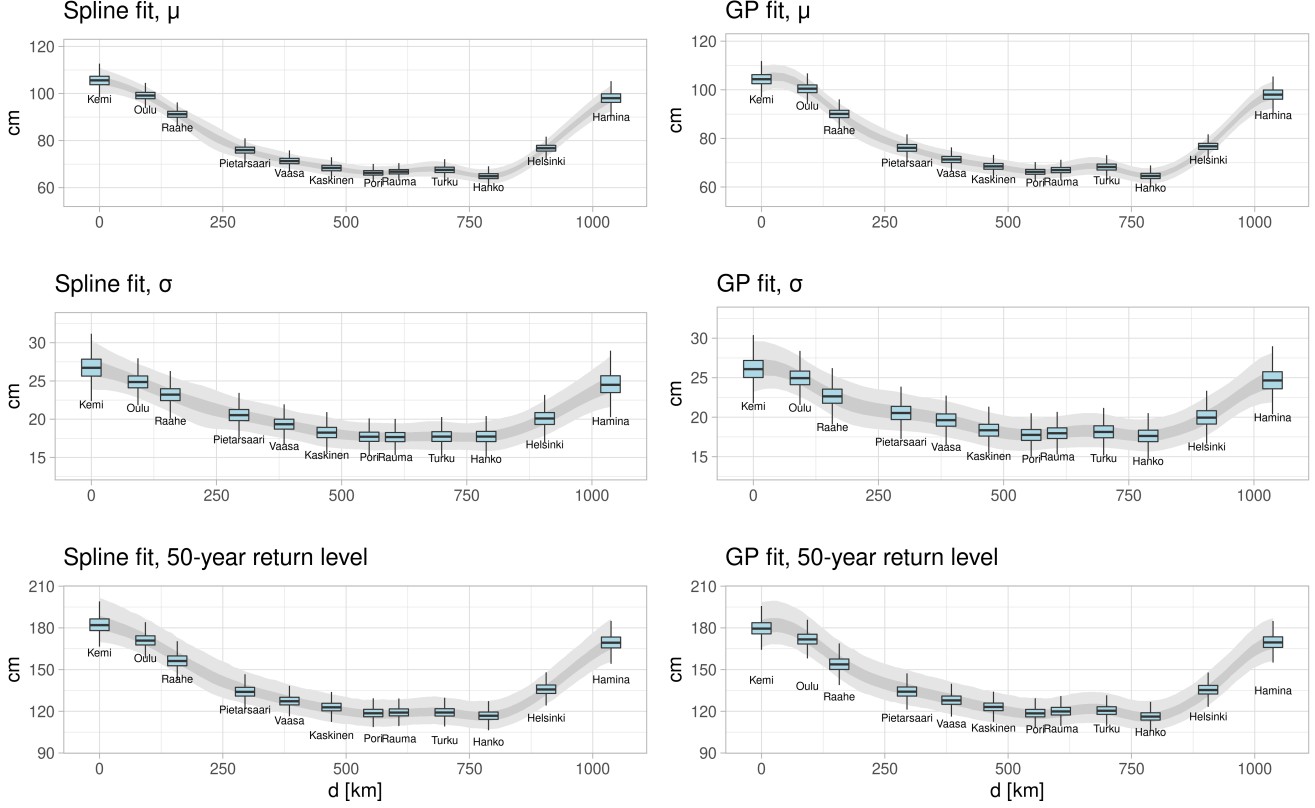

**Figure 2.** Illustration of the (left) spline and (right) Gaussian process fits for the (top) location and (middle) scale parameter with respect to the distance from Kemi tide gauge. Darker (light) shading denotes the interquartile (95 %) range of the parameter estimates. The figure also shows box-plots of the corresponding parameters at the tide gauge locations. The bottom row shows similar plots for the 50-year return level of annual maximum sea levels for both models. The box covers the interquartile range (IQR) and the median value is highlighted by the horizontal line. The length of the whiskers is one and half times the IQR.

of the distance dependence is similar for both the Spline and GP model. Note that we do not attempt to extrapolate beyond Kemi or Hamina, as the resulting parameter estimates would be highly uncertain. The bottom row in Fig. 2 illustrates how the spatial 50-year return level estimates look like for both models. The shape parameter $\xi$ has been sampled from the joint
posterior distribution of the tide gauge specific parameter values when drawing these plots. It is seen that the spatial return level estimates vary smoothly along the coast and match relatively closely with the tide gauge specific estimates of these models. In the following, we concentrate on the tide gauge specific estimates, as it allows us to compare the results of all four models.

## 4  Results

We first evaluate the goodness-of-fit of the hierarchical models against the observations and compare their performance against
the tide gauge specific GEV fits. We then have a more in-depth look at the simulated posterior parameter distributions for
the hierarchical models and illustrate how the return level estimates differ between Separate and the hierarchical models. We
also briefly discuss whether some theoretical limits for the annual maximum sea level could be inferred from the estimated
parameters.

### 4.1  Evaluation of model fit against observations

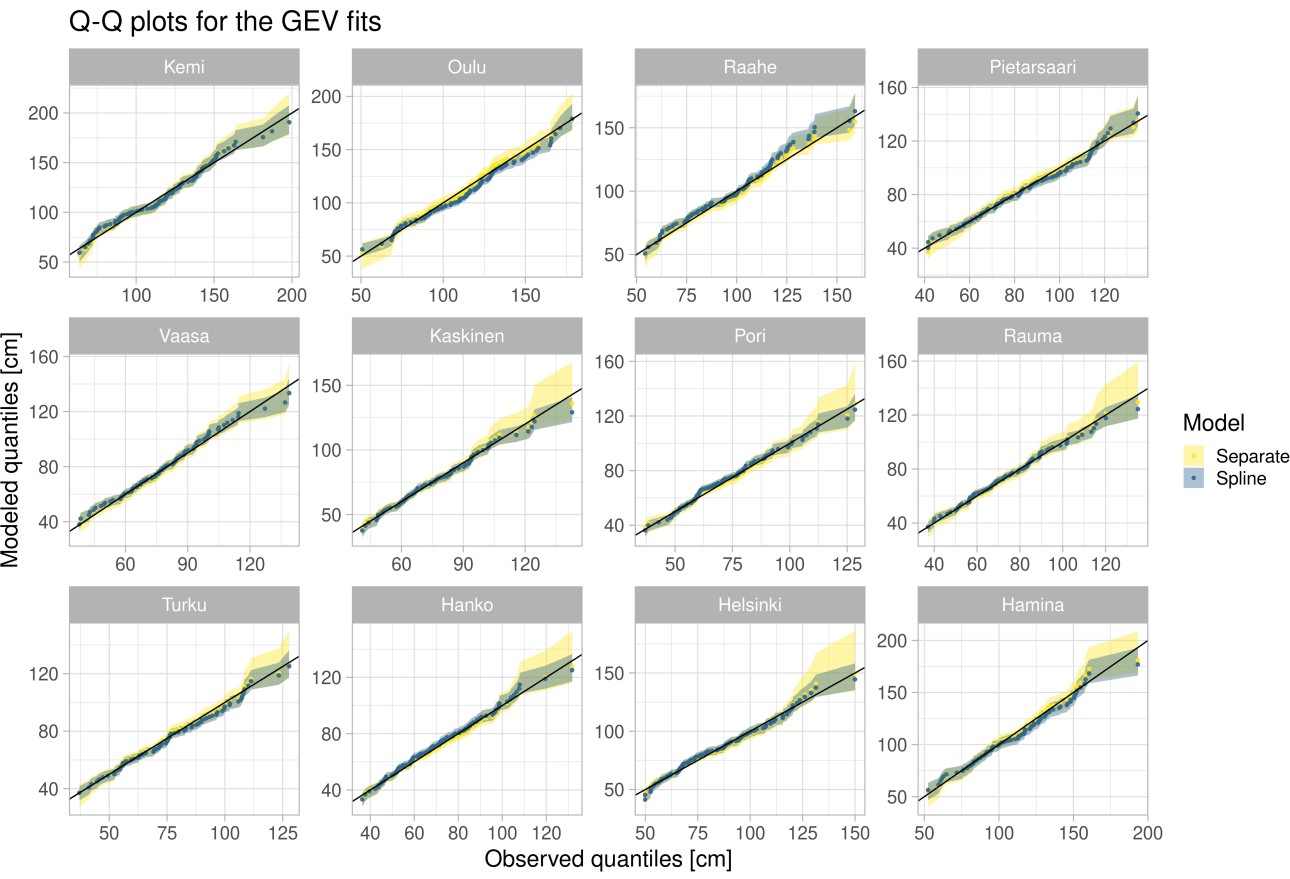

**Figure 3.** Quantile-quantile plots showing the modeled quantiles for tide gauge specific fits (Separate) and for Spline when plotted against
the observed quantiles. Median values are given as points and the shading shows the 95% uncertainty range for the modeled quantiles. The
diagonal line is also shown in each panel.

To obtain some insights on model adequacy and the goodness of the separate and hierarchical model fits, quantile-quantile
(Q-Q) plots calculated against the observed return levels are shown in Fig. 3. As the Q-Q plots are very similar for all hierar-

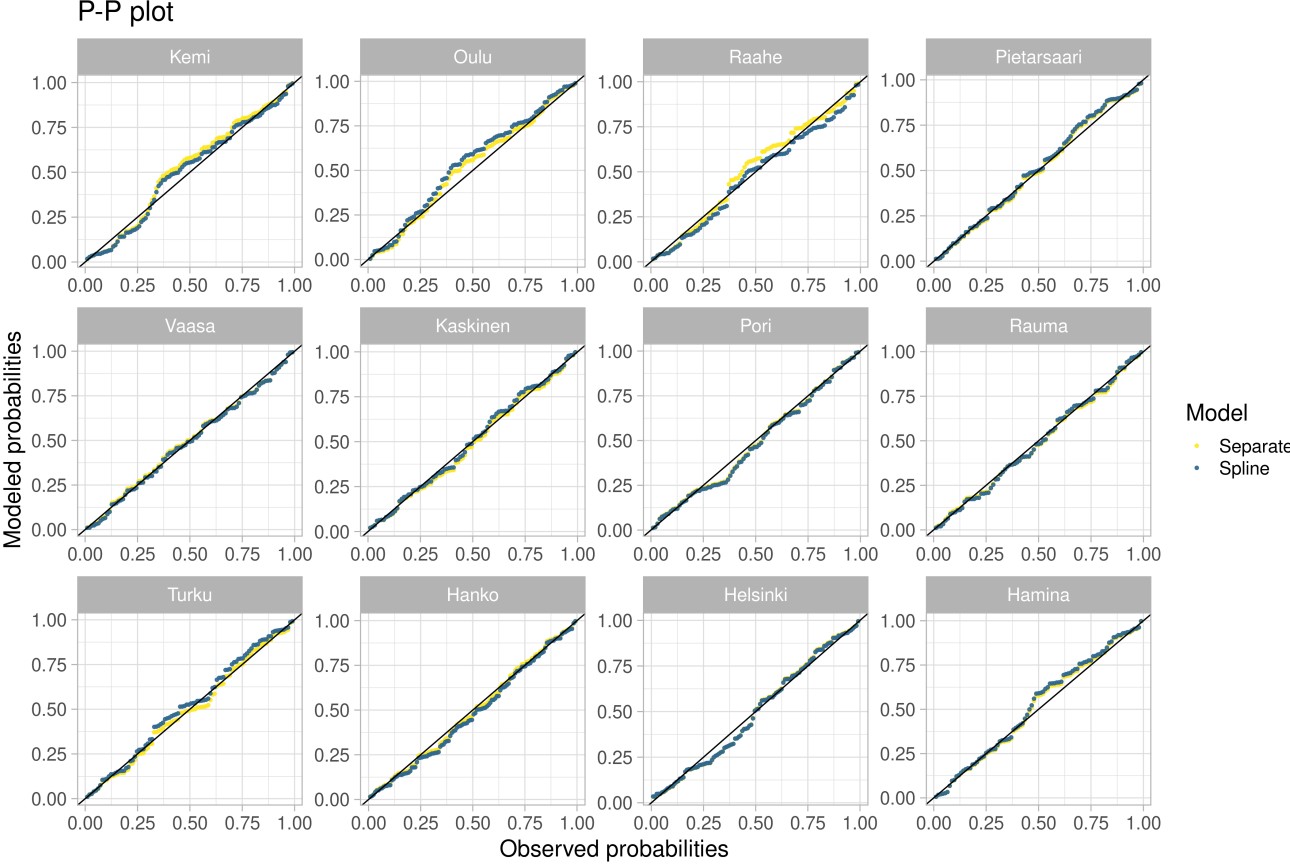

**Figure 4.** Probability-probability plots calculated using median values of the GEV parameters for tide gauge specific fits (Separate) and for Spline when plotted against the observed probabilities. The diagonal line is also shown in each panel.

chical models, the results are shown only for the Spline model. Visual inspection of the tide gauge specific panels shows that both GEV models fit reasonably well to the data in most tide gauges, although the observational estimates tend to be outside the 95% uncertainty range in some cases. Fig. 3 also shows that although the median results are close to each other for these models, the uncertainty range is indeed larger for Separate in many locations. Fig. 4 shows the probability-probability (P-P) plots for the same models. While the panels do not reveal any major discrepancies from the observed probabilities in most tide gauges, the match is less good for those tide gauges that are located close to the ends of the Bay of Bothnia and Gulf of Finland. In particular, the fit seems to be less good for Spline compared to Separate in Oulu. Nevertheless, both Q-Q and P-P plots indicate that, with some tide gauge specific discrepancies with the observations, the models provide reasonable fits to the data.

## 4.2 Leave-one-out cross validation

To compare the relative performance of the GEV models, we performed leave-one-out cross validation with Pareto smoothed importance sampling (PSIS-LOO) implemented in the `loo` package in R (Vehtari et al., 2017, 2020). `loo` provides the expected logarithmic point-wise predictive density $\mathrm{elpd}_{\mathrm{loo}}$ for the out-of-sample predictions and also estimates the effective number of parameters $p_{\mathrm{loo}}$. While Bayesian LOO cross-validation has some limitations for model selection purposes (e.g., see discussion in Gronau and Wagenmakers (2019a) and Gronau and Wagenmakers (2019b)), it is nevertheless useful in our case for highlighting possible differences in the model performance.

| | $\Delta\mathrm{elpd}_{\mathrm{loo}}$ | $p_{\mathrm{loo}}$ |
|---|---|---|
| Spline | 0.0 (0.0) | 16.4 (1.2) |
| GP | -1.2 (1.0) | 18.5 (1.2) |
| Common | -5.2 (2.1) | 22.7 (1.4) |
| Separate | -9.8 (3.3) | 29.2 (1.9) |

**Table 1.** Leave-one-out cross validation statistics for the implemented GEV models. The first column shows the difference in the expected logarithmic point-wise predictive density ($\mathrm{elpd}_{\mathrm{loo}}$) with respect to the best performing model and the second column the effective number of parameters. Values within the brackets are the standard errors for the estimated statistics.

The LOO cross-validation statistics are listed in Table 1. The first column shows the difference in $\mathrm{elpd}_{\mathrm{loo}}$ with respect to the best model for which this value is zero, together with its standard error estimate. Spline has the highest $\mathrm{elpd}_{\mathrm{loo}}$ value out of all models with GP having only slightly worse results. Separate has the smallest $\mathrm{elpd}_{\mathrm{loo}}$, which indicates that the hierarchical models might provide a better fit to the observed annual extremes. The second column shows the effective number of parameters ($p_{\mathrm{loo}}$) together with the estimated standard error, whose value gives a rough estimate of model complexity. In line with $\Delta\mathrm{elpd}_{\mathrm{loo}}$ Spline has the smallest $p_{\mathrm{loo}}$ which is roughly half of that for Separate.

To assess the performance of the spatial models in ungauged locations, we performed an additional experiment in which the tide gauges were left out one at time before fitting Spline and GP to the observations and the obtained fit was used to estimate the 50-year return level in the omitted tide gauge location. This procedure was repeated over all tide gauges apart from Kemi and Hamina, as our models are not suitable for extrapolating the results spatially. We then calculated absolute and relative bias and mean absolute error (MAE) for the posterior median and conditional rank probability score (CRPS; Hersbach, 2000) for the full posterior distribution against the observed 50-year return level. The observed 50-year return level has been estimated by first calculating the exceedance probability for the observed annual maxima based on Weibull plotting positions and then interpolating the estimated probabilities between the observed values. The resulting statistics are shown in Table 2, when averaged over the ten tide gauges. The spatial models fitted without the target tide gauge ($\mathrm{Spline}_{\mathrm{loo}}$ and $\mathrm{GP}_{\mathrm{loo}}$) have worse statistics than the "full" models except for the model bias for $\mathrm{Spline}_{\mathrm{loo}}$. In particular, $\mathrm{GP}_{\mathrm{loo}}$ has the largest errors apart from MAE. However, absolute differences in the error statistics to the model estimates based on full data set are not large, which suggests that both Spline and GP are able to provide useful posterior predictions in ungauged locations.

|  | Separate | Common | Spline | GP | Spline$_{\text{loo}}$ | GP$_{\text{loo}}$ |
|---|---|---|---|---|---|---|
| Bias (cm) | 0.4 | -1.2 | -1.4 | -1.4 | -0.7 | 2.6 |
| Rel. bias (%) | 0.4 | -0.9 | -1.1 | -1.0 | -0.5 | 2.3 |
| MAE (cm) | 4.5 | 3.7 | 3.7 | 3.6 | 6.8 | 5.8 |
| CRPS (cm) | 3.1 | 2.7 | 2.8 | 2.6 | 4.4 | 4.4 |

**Table 2.** Absolute and relative bias, mean absolute error (MAE) and conditional rank probability score (CRPS) calculated with respect to the observed 50-year return level when averaged over the tide gauges apart from Kemi and Hamina. The statistics are shown for the four models fitted using all tide gauge records and (last two columns) for Spline and GP, when the target tide gauge has been left out data before fitting the models.

### 4.3 Posterior parameter distributions

We next take a look at the posterior parameter estimates obtained with MCMC. To first illustrate some of the properties of the posterior parameter distributions, bi-variate parameter distributions are shown in Fig. 5 for Separate and Spline in the Kemi tide gauge. The panels in this plot show that $\mu$ and $\sigma$ are negatively correlated with $\xi$ to a certain degree for Separate, as the location

and scale parameters tend to increase for more negative shape parameter values. The parameters are less strongly correlated for the Spline model and the distribution shape is more Gaussian due to the effect of pooling. The bi-variate distributions look well identified for both models but for Spline cover visibly narrower parameter ranges compared to Separate. Very similar results are obtained for the other hierarchical models and therefore, are not shown here.

Figure 6 shows the posterior parameter distributions for all models. Both the location $\mu$ and scale $\sigma$ parameter generally

increase towards the ends of the two bays. The location parameter $\mu$ obtains its largest values in Kemi, Oulu and Hamina tide gauges, where the median value for $\mu$ approaches or even exceeds 100 cm. The posterior median for $\sigma$ is close to, or exceeds, 25 cm in the same tide gauges. Furthermore, the spread of the posterior parameter distribution tends to be largest in these tide gauges for both $\mu$ and $\sigma$. Note that the posterior parameter uncertainty in $\sigma$ tends to be somewhat larger for Separate compared to the hierarchical models. Variations in $\mu$ across the tide gauges are similar for all models, but for the scale parameter $\sigma$,

Separate has slightly larger overall variations compared to the hierarchical models. This could be due to the shrinkage effect, which tends to bring the station specific parameter values of the hierarchical models towards the overall mean.

In contrast with $\mu$ and $\sigma$, the shape parameter $\xi$ does not have as clear connection with the distance from the two bays. The overall posterior median for $\xi$ is around -0.16 for all models. However, for Separate the posterior median tends to be more negative around the coast of the Bothnian Bay and less negative to south from Vaasa on the coast of the Bothnian Sea.

For the hierarchical models the shape parameter values vary only weakly with the tide gauge location, as this is parameter is in our hierarchical models formulation less sensitive to location specific aspects of data. Even when the uncertainty in the posterior parameter estimates is taken into account, $\xi$ is consistently negative for all hierarchical models, although for Separate the posterior distribution of $\xi$ has a long tail towards positive values in some locations. The typically negative shape parameter value for all models suggests that annual maximum sea levels follow a 3-parameter Weibull distribution on the

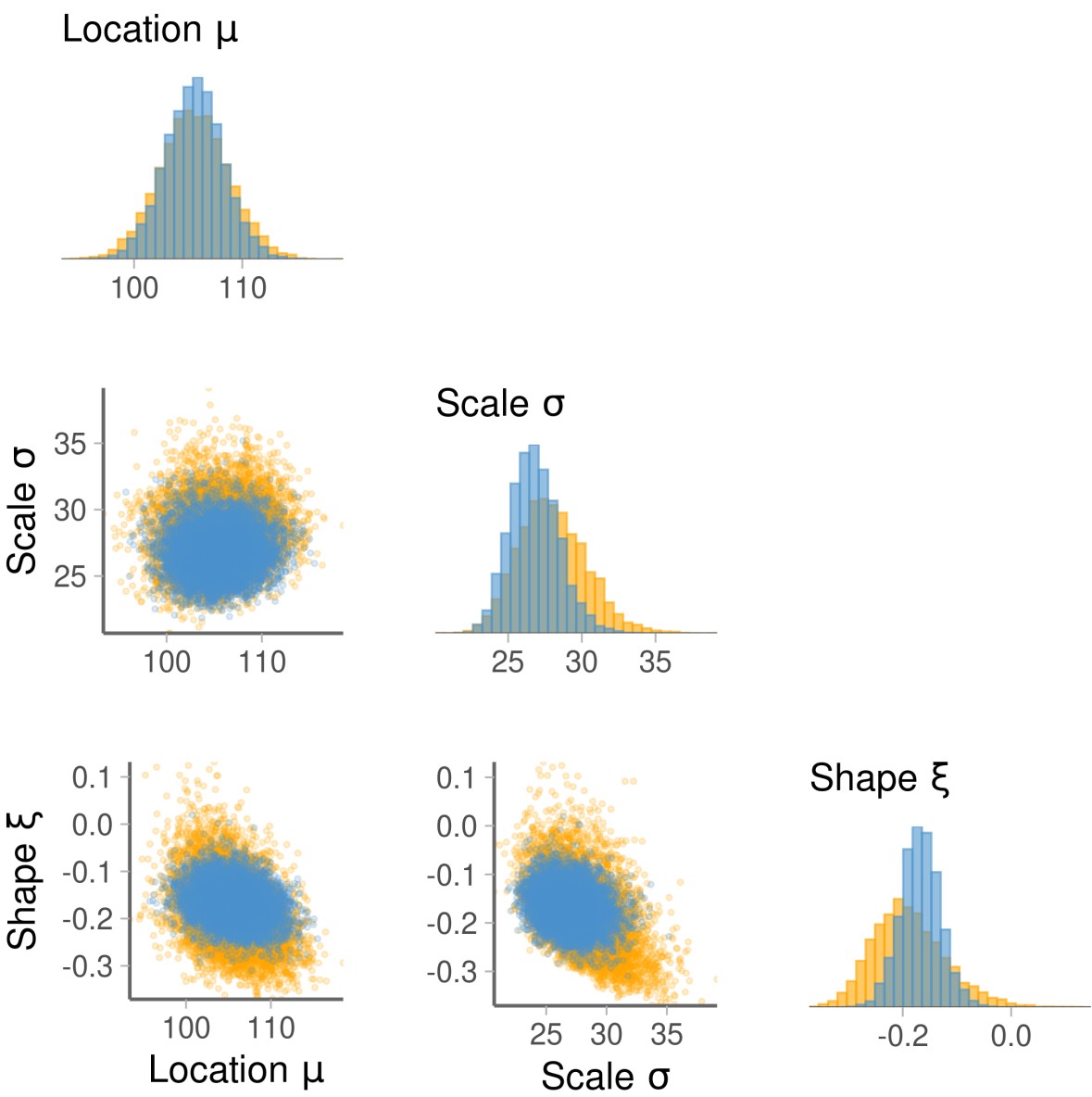

**Figure 5.** Example of posterior distributions for the GEV parameters in Kemi tide gauge obtained with (orange) Separate and (blue) Spline, respectively. Histograms of the three GEV parameters are shown on the diagonal and the corresponding bi-variate scatter plots on the lower triangle. The location and scale parameter values are given in centimeters.

Finnish coast. This result is in line with the previous study by Marcos and Woodworth (2017), which also suggested a negative shape parameter for the Finnish and the neighbouring tide gauges. A slightly contrasting result was obtained by Soomere et al. (2018), who estimated shape parameter values close to zero from an ocean model output along the Estonian coast. Furthermore,

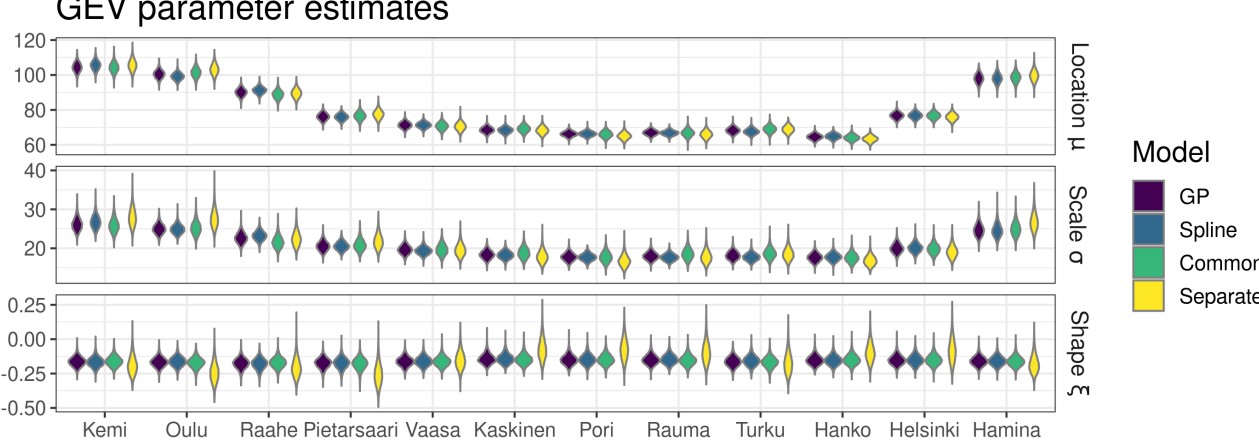

**Figure 6.** Posterior parameter distributions for all the tested models at each tide gauge. The location and scale parameter values are given in centimeters.

in their additional analysis based on tide gauge observations the shape parameter varied strongly depending on location and method used to estimate the GEV distribution parameters. We note that as the behavior of annual sea level extremes is likely different on the Estonian side of Gulf of Finland, their results are not directly comparable to ours.

When $\xi < 0$, the return level $y_p$ has an upper limit at $y_0 = \mu - \sigma/\xi$, which could be used, at least in theory, to estimate the highest possible value that the annual maximum sea level can reach along the Finnish coast. We illustrate the upper limit estimates in the next section.

To conclude, compared to the separate fits, the hierarchical models have a narrower uncertainty range for the scale and shape parameter. Overall, posterior parameter distributions are very similar for the hierarchical models regardless of the tide gauge. This shows that pooling information across the tide gauges narrows the uncertainty range in the posterior parameter estimates.

### 4.4 Posterior predictive simulations

The estimated GEV parameters shown in the previous section can be used to calculate theoretical estimates for the N-year return levels on tide gauge locations. The estimated return levels are illustrated for all models and for two return periods in Fig. 7. The upper panel shows box-plots of 50-year return levels together with the return level estimated from the observations. In most cases, the observed return level matches relatively well with the model estimates. One must keep in mind that due to the relatively short length of the available time series, the observed return level estimates are uncertain and likely affected by sampling variability.

The lower panel in Fig. 7 shows estimates of much rarer, 1000-year return levels. In this case, differences between Separate and the hierarchical models have become noticeably more visible compared to the 50-year return level. The median values are slightly higher for the hierarchical models on tide gauges from Oulu to Pietarsaari, while Separate has markedly higher median

## 50-year return level

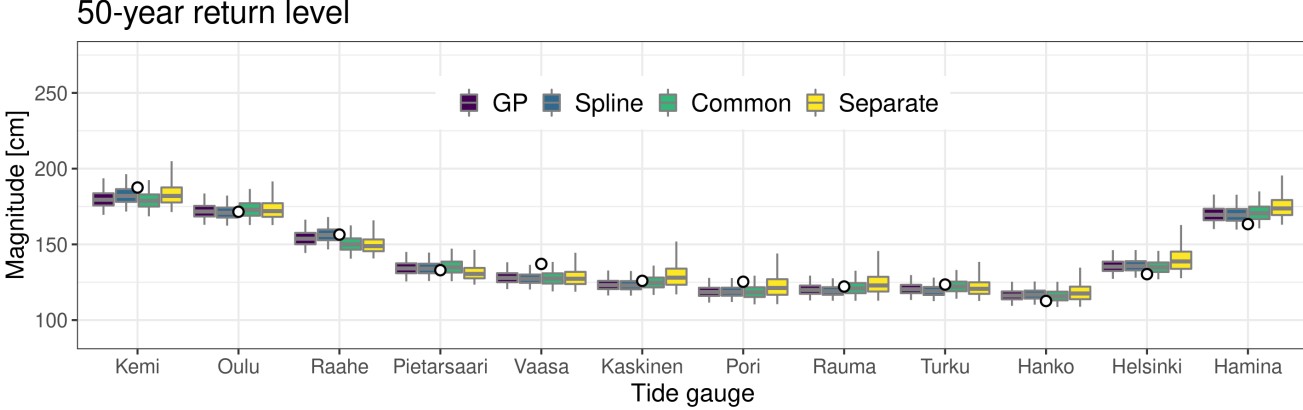

## 1000-year return level

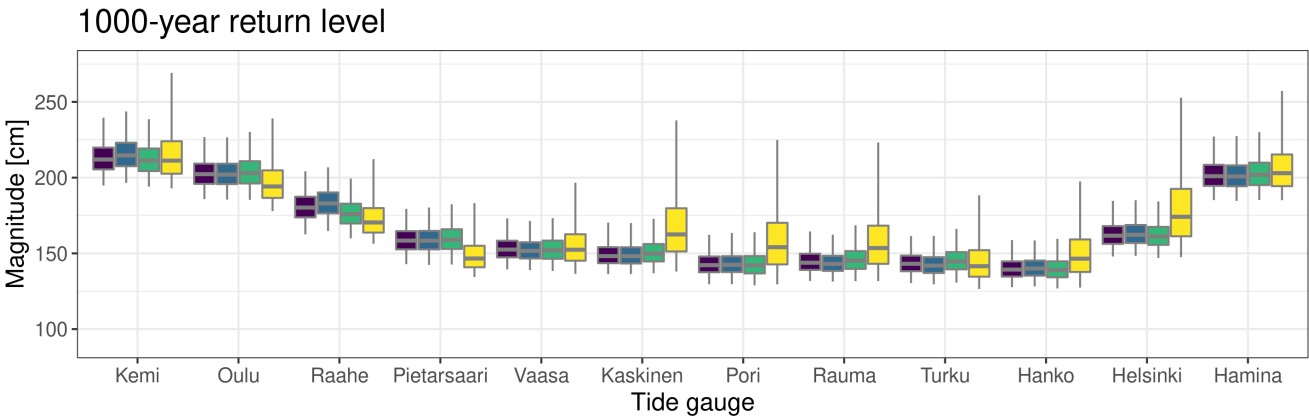

**Figure 7.** Example plots showing the posterior predictions for (top) 50-year and (bottom) 1000-year return levels for all four models. For comparison, the upper panel contains (circles) the observed 50-year return levels calculated from the observations. The boxes cover the inter-quartile range and are shown together with the median value, while the whiskers extend over the 95% percentile range.

.

values in many tide gauges from Vaasa to Hamina. For Separate, the 95 % uncertainty range in the return level estimates tends to be smaller (larger) in the former (latter) locations, although the uncertainty range is in absolute terms largest for this model in all tide gauges. The hierarchical models predict very similar 1000-year return level values and also have similar uncertainty ranges in all locations. Most of the features in the return level estimates obtained with Separate are directly related to the larger uncertainty in the estimated shape parameter values, which also exhibit larger spatial variations for this model than for the hierarchical models.

To further illustrate how much the hierarchical modeling approach reduces the prediction uncertainty, Table 3 shows the standard deviation of the predicted 50-year return level in the tide gauge locations and its ratio (in percentage) with respect to Separate for the hierarchical models. The spread is reduced in all cases and in some locations for Spline and GP is less than 50 percent of that for Separate. There are no major differences between the hierarchical models, although the reduction in the predictive uncertainty tends to be slightly smaller for Common than for the two spatial models. This supports our conclusion that the hierarchical models are able to reduce uncertainty in the posterior predictions.

| Model | Kemi | Oulu | Raahe | Pietarsaari | Vaasa | Kaskinen | Pori | Rauma | Turku | Hanko | Helsinki | Hamina |
|---|---|---|---|---|---|---|---|---|---|---|---|---|
| Separate | 85 | 75 | 66 | 59 | 67 | 91 | 86 | 85 | 67 | 67 | 92 | 85 |
| Common | 61 (72) | 62 (82) | 56 (84) | 55 (93) | 50 (75) | 49 (55) | 49 (57) | 51 (60) | 48 (72) | 43 (65) | 48 (52) | 62 (74) |
| Spline | 63 (74) | 50 (66) | 54 (82) | 48 (82) | 42 (63) | 41 (45) | 40 (47) | 38 (44) | 40 (60) | 39 (59) | 47 (51) | 59 (70) |
| GP | 61 (71) | 53 (70) | 56 (84) | 50 (85) | 45 (68) | 42 (47) | 42 (49) | 42 (49) | 42 (63) | 41 (61) | 48 (52) | 58 (69) |

**Table 3.** Standard deviation of the predictive distribution for the 50-year return level (in mm), shown separately for each tide gauge and model. The percentage fraction of standard deviation with respect Separate is given for the three hierarchical models in the brackets.

As was mentioned in Sect. 2, individual extremely high sea level observations might markedly influence model fitting. To check this, we performed a sensitivity test in which the observation from January 1984 was left out of the time series before fitting the models to the data. Removal of this observation lead to a slightly more negative shape parameter value for Separate especially in Kaskinen and Pori, which reduced its median estimate of 1000-year return level closer to the hierarchical models in these locations (not shown). Return level estimates for the hierarchical models were only slightly changed in this test, which highlights higher sensitivity of separate model fits to anomalous observations. We note that the 1000-year return level estimates require extrapolation to such cases that are not present in the observational records. Therefore, the aforementioned results should be interpreted cautiously.

As discussed in the previous section, the mostly negative shape parameter suggests that, in theory, we could infer an upper limit the sea level can reach given the assumptions on data and models are met. As a sanity check for our models, we briefly illustrate how high values the hierarchical models give at the theoretical upper limit. We stress that the shown values should not be interpreted as actual limits for the sea level but, rather, as a hypothetical result provided by the hierarchical models.

The distribution of $\mu - \sigma/\xi$ is illustrated for the hierarchical models in Kemi tide gauge in Fig. 8. All distributions look very similar and are positively skewed towards larger values. The median value is almost identical for all models and exceeds 260 cm. These values are 63–65 cm higher than the largest value in the observed time series which sounds reasonable, although the very long right tail highlights the substantial uncertainty associated with the upper level estimates.

The theoretical upper limit estimates are summarised for the rest of the tide gauges in Table 4. Unsurprisingly, the median estimates are highest at those tide gauges, which are located closest to the end of the Bothnian Bay and Gulf of Finland, where they exceed 250 cm. As in the Kemi example, all hierarchical models give very similar results. Overall, differences between the highest observed sea levels and the theoretical upper limit vary between 47–73 cm, depending on the location. Interestingly,

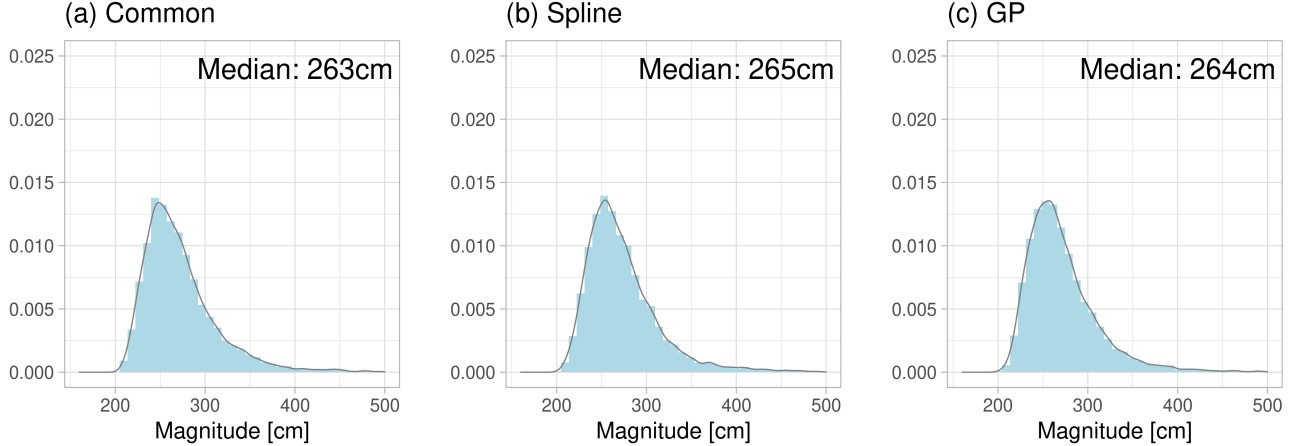

**Figure 8.** Theoretical upper limit ($\mu - \sigma/\xi$) for the annual maximum sea level in Kemi tide gauge inferred from the posterior parameter distributions. The results are shown for the hierarchical models only.

there is also a local maxima in the upper limit estimates in Kaskinen, which is similar to the feature that was seen in the
1000-year return level estimates for Separate in Fig. 7.

| Tide gauge | Common | Common-obs | Spline | Spline-obs | GP | GP-obs |
|---|---|---|---|---|---|---|
| Kemi | 263 (220–393) | 65 | 265 (221–393) | 67 | 264 (221–385) | 65 |
| Oulu | 248 (205–357) | 70 | 251 (208–369) | 73 | 249 (207–360) | 70 |
| Raahe | 215 (178–310) | 56 | 221 (179–312) | 62 | 218 (179–311) | 59 |
| Pietarsaari | 193 (156–284) | 59 | 194 (157–289) | 60 | 194 (157–275) | 59 |
| Vaasa | 191 (158–289) | 52 | 191 (158–284) | 52 | 192 (158–284) | 53 |
| Kaskinen | 194 (160–338) | 51 | 195 (160–353) | 52 | 193 (160–342) | 51 |
| Pori | 184 (149–314) | 56 | 184 (149–323) | 56 | 183 (149–308) | 55 |
| Rauma | 186 (152–307) | 51 | 186 (152–305) | 50 | 185 (153–308) | 50 |
| Turku | 180 (147–268) | 52 | 180 (146–286) | 52 | 179 (147–267) | 51 |
| Hanko | 179 (147–303) | 48 | 180 (147–283) | 48 | 179 (148–284) | 47 |
| Helsinki | 206 (168–347) | 56 | 208 (170–350) | 58 | 206 (170–335) | 56 |
| Hamina | 252 (210–381) | 58 | 252 (211–377) | 59 | 252 (211–381) | 59 |

**Table 4.** Median value of the theoretical upper limit ($\mu - \sigma/\xi$) of the annual maximum sea level in units of centimeters, calculated separately for each tide gauge. The 95% credible interval is shown within the brackets. Differences to the highest observed annual sea levels are also provided for each model.

## 5 Discussion

To put the obtained results into a broader perspective, we briefly compare our results to those available in the literature from both statistical and dynamical model studies. We stress that this comparison is qualitative at the best, as it is difficult to make direct comparisons due to differences in the used methods and observations.

Wolski et al. (2014) provided 50-year return level estimates based on stationary GEV fit to observations from 1960–2010 for Helsinki (163.8 cm) and Kemi (209.8 cm) tide gauges. These results are 25.1–31.1 cm higher than the median values for our models and closer to the estimated 1000-year return levels. This is likely due to differences in the study period and methods used. Using idealised cyclone winds as forcing in a hydrodynamic ocean model, Averkiev and Klevannyy (2010) estimated how high sea levels could be associated with strong low pressure systems in the Gulf of Finland. Their results (184 cm in

Hanko and 186 cm in Helsinki) are higher than our 1000-year return level estimates for the hierarchical models and closer to their median upper limit estimates. Särkkä et al. (2017) performed an 850-year long simulation with a hydrodynamic ocean model using regional climate model simulations as input. In their results, sea level could reach 225 cm height in Helsinki, which is somewhat higher, but still similar to our median upper limit estimate in this location.

    As a deliberate choice in our study, long-term changes in mean sea level were excluded from the analysis by detrending the

time series. Inclusion of mean sea level would be required, if we were to assess the overall flooding risk related to sea level extremes both in the present and future climate, as mean sea level has been a major driver of sea level extremes also in the Finnish coastal region (e.g., Marcos and Woodworth, 2017). Furthermore, studies have suggested that future changes in the sea level extremes are strongly associated with changes in mean sea level in the Baltic Sea region (e.g., Meier et al., 2004; Gräwe and Burchard, 2012; Vousdoukas et al., 2016, 2017). In Finland, changing mean sea level is expected to increase flooding risk

along the coast of Gulf of Finland and the Gulf of Bothnia, whereas post-glacial land uplift is likely to counter most of changes in the mean sea level in the Bothnian Bay (Pellikka et al., 2018).

    We have used GEV distribution to model the overall annual sea level maxima (after removing long-term trends). However, different processes contribute to variations in sea level in the Baltic Sea and cause it to fluctuate at different temporal scales. Therefore, alternative modeling approaches could be considered to model separately the different sea level fluctuations and

their contributions. For example, Soomere et al. (2015) have used an approach in which separate statistical models were fitted to weekly-scale and local storm-surge driven sea level fluctuations.

    We also recognise some limitations in our study. Firstly, it was assumed that the time series of sea level extremes are stationary after they have been detrended. This assumption is unlikely to be completely true. While not giving a rigorous conclusion, visual inspection of the time series in Fig. 1 suggests that changes might have occurred both in the level and

variability of annual maxima. Furthermore, studies have indicated that sea level extremes have exhibited variations in the Baltic Sea region over different time scales (Johansson et al., 2001; Ribeiro et al., 2014; Marcos and Woodworth, 2017; Kudryavtseva et al., 2021) and that these variations were associated with variations in large scale atmospheric conditions (Samuelsson and Stigebrandt, 1996; Johansson et al., 2001; Ribeiro et al., 2014; Marcos and Woodworth, 2017; Kudryavtseva et al., 2021; Passaro et al., 2021). As with changing mean sea level, future climate is also expected to bring changes in storm surges in

the Baltic Sea region (e.g., Vousdoukas et al., 2016, 2017, 2018). Our models do not account for non-stationarity in extremes related to short-term sea level variations and cannot be used to assess such changes.

One remedy would be to model temporal dependence directly by allowing (e.g.) linear trends in the GEV parameters (e.g., Ribeiro et al., 2014; Marcos and Woodworth, 2017; Kudryavtseva et al., 2021). Physical covariates, which describe large scale atmospheric circulation conditions around the Baltic Sea region, could also be used to account for interannual-to-decadal scale
variations in the annual maximum sea levels. For example, Marcos and Woodworth (2017) assessed connections between sea level extremes and North Atlantic Oscillation (NAO) index in the North Atlantic region. Their results showed that even after taking the effect of mean sea level into account, NAO explained part of temporal variations in the sea level extremes in the Baltic Sea. We aim to incorporate such physical covariates in our models in the following studies.

Another limitation for our hierarchical models is that they only account for dependence in the marginal GEV parameters
and do not take additional residual dependence (dependence in annual maxima between different tide gauges) into account. Exclusion of residual dependence implies that our uncertainty estimates are likely slightly too narrow. One way to address this shortcoming is provided by Calafat and Marcos (2020), who use a max-stable process to capture the residual dependence. Their approach is, however, outside the scope of this paper.

## 6   Conclusions

In this study, Bayesian hierarchical modeling was applied to estimate theoretical return levels from annual maximum sea level on tide gauges located along the Finnish coast. Three hierarchical descriptions of the parameters of the generalised extreme value (GEV) distribution were compared against individual fits at twelve tide gauges. The main motivation was to test how the hierarchical description of the distribution parameters affects the estimation of, and uncertainty in, them and the corresponding estimates of return levels. The simplest hierarchical model assumes that station specific distribution parameters come from
the same joint Gaussian hyper-distributions. In addition, two hierarchical model structures based on B-splines and Gaussian processes were implemented in which the distance with respect to Kemi tide gauge along the Finnish coast was used as a covariate in the model formulations. These two models also allow, in principle, the estimation of return levels between the tide gauges.

The main results obtained from subsequent analysis are:

– All hierarchical models provide similar results for the GEV parameters and reduce the posterior parameter uncertainty in comparison to the individual model fits. In addition, the shape parameter is consistently negative on the Finnish coast with median values around -0.16 for all models.

– Examples for the return level estimates show that the 50-year return level is well captured by the hierarchical models. For rarer (1000-year) events, individual fits tend to have a larger uncertainty range and they also give higher return level
estimates compared to the hierarchical models in most locations.

– Median values of the theoretical upper limits computed with the hierarchical models indicate differences of 47–73 cm in comparison to the observed maximum sea levels in the observational time series. However, uncertainty in these estimates is large and therefore, they should be interpreted with extreme caution.

While the results suggest that our hierarchical models provide an improvement over separate fits to tide gauge time series, the study is region specific and exploits regional geographical features. For regions with shorter tide gauges records available for analysis, it is expected that the hierarchical modeling approach has larger benefits in comparison to tide gauge specific models. Furthermore, other modeling approaches should be included along our approach and compared with the local sea level observations to find the best solutions. One must also keep in mind that the results still contain uncertainties due to possibly missing components (long-term temporal evolution, physical information) in the model formulation and the limited sample size for the observed extremes. To conclude, care needs to be taken when interpreting the above-mentioned results. Improvements for the tested models, e.g., by including missing non-stationary components, will be addressed in future studies.

*Code and data availability.* Code for reproducing this study has been made publicly available in ZENODO https://doi.org/10.5281/zenodo.7838345 (Räty and Laine, 2023). The package contains the implemented Stan models, R scripts for running the models and for reproducing most of the analysis made in this study. The detrended tide gauge time series of annual maximum sea levels required by the scripts are also available in ZENODO https://doi.org/10.5281/zenodo.5807461 (Räty and Johansson, 2021).

*Author contributions.* All authors contributed to the design of the study. MJ and JS provided the tide gauge data and OR pre-processed the data with the help of ML, JS and MJ. OR and ML implemented the statistical models and conducted the analysis presented in the paper. OR, UL and ML wrote the manuscript with critical comments from the other authors.

*Competing interests.* The authors declare that there are no competing interest regarding this study.

*Acknowledgements.* The authors thank the two anonymous reviewers for their constructive comments, which helped to improve the manuscript. This research was funded by the Finnish State Nuclear Waste Management Fund (VYR) through SAFIR2022 (the Finnish Research Programme on Nuclear Power Plant Safety 2019-2022) and the Finnish Academy Funded Advanced data fusion methods for environmental modeling (ADAFUME) project (decision number 321890).

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
