# Peer review of "Bayesian hierarchical modeling of sea level extremes in the Finnish coastal region"

_Natural Hazards and Earth System Sciences, 2021_

## Author Comment (AC1)

**Reply to Reviewer 1**

We would like to thank the reviewer for his/her insightful comments on our manuscript which have helped us to improve it. In the following, the original reviewer comments are given in italics and our point-by-point responses to the reviewer's comments in roman font with planned changes to the text put into quotation marks.

*This paper presents a spatial Bayesian hierarchical model of sea-level extremes and uses it to analyse tide gauge observations along the Finish coastline. Estimates of extreme sea-level event probabilities, which typically are expressed in terms of return levels, are crucial to flood risk quantification. However, such estimates are often subject to large uncertainty owing to issues related to the small sample sizes and large data dispersion typical of tide gauge observations. Furthermore, when using traditional single-site approaches, estimates of event probabilities are only possible at gauged locations. These issues can be partly overcome by exploiting spatial dependencies in extreme sea levels, or simply by pooling information across data sites, which leads to estimates of return levels with reduced uncertainty and allows for estimation at unobserved locations. Despite the advantages of spatial modelling, most studies of sea-level extremes to date have analyses extremes on a site-by-site basis. In this regard, this paper represents a valuable contribution to the literature on sea-level extremes. The paper shows that pooling information across space leads to more robust estimates of event probabilities, though in this study all tide gauge records are relatively long and as a result the single-site model ('Separate') is still able to estimate the GEV parameters with high confidence. The benefits of spatial modelling are much larger in regions with short tide gauge records, and this should be more strongly emphasized in the paper. The paper is well written, the methods are valid, and overall the results are interesting. I do not have any major objections to the paper, but I do have some comments and suggestions, as outlined below, that would like to see addressed before the paper is published in NHESS.*

Thank you for the supportive comments. We will include the following sentence in the Conclusions section to better emphasize the larger benefit of hierarchical modeling approach in regions with shorter tide gauge records:

"...For regions with shorter tide gauges records available for analysis, it is expected that the hierarchical modeling approach has even larger benefits in comparison to tide gauge specific models."

*General comments:*

1. *One of the motivations for using spatial modeling is the ability to make estimates at ungauged locations. However, other than in Fig. 2, the paper focuses on estimates at gauged sites and does not sufficiently assess the skill of the Bayesian models at ungauged sites. I would suggest the authors perform an experiment in which they leave one tide gauge out at a time, estimate the GEV parameters at the omitted site, and then compare the result with estimates based on all the data. I would also suggest the authors include a map of gridded estimates of 50-year return levels along the Finnish coastline.*

As suggested by the reviewer, we performed an additional test with Spline and GP in which each tide gauge record (apart from Kemi and Hamina, as we did not want to extrapolate spatially) was left out one at a time before fitting the models. We decided to look at the posterior predictions and calculated the 50-year return level at the omitted locations. The predicted return levels were evaluated by calculating bias, mean absolute error and continuous ranked probability score against the return level estimated from observations. We will add a new table (Table 2) showing the statistics when averaged over the omitted tide gauges and some discussion on these results to the manuscript as follows:

|            | Separate | Common | Spline | GP  | Spline$_{LOO}$ | GP$_{LOO}$ |
|------------|----------|--------|--------|-----|----------------|------------|
| CRPS (cm)  | 3.2      | 2.8    | 3.0    | 2.8 | 4.6            | 6.2        |
| Bias (cm)  | 4.9      | 3.3    | 3.1    | 3.2 | 3.9            | 7.2        |
| MAE (cm)   | 5.7      | 4.2    | 4.2    | 4.1 | 6.3            | 8.4        |

**Table 2.** Bias, mean absolute error (MAE) and conditional rank probability score (CRPS) calculated with respect to the observed 50-year return level when averaged over the tide gauges apart from Kemi and Hamina. The statistics are shown for the four models fitted using all tide gauge records and (last two columns) for Spline and GP, when the target tide gauge has been left out data before fitting the models.

"To assess the performance of the spatial models in ungauged locations, we performed an additional experiment in which the tide gauges were left out one at time before fitting Spline and GP to the observations and the obtained fit was used to estimate the 50-year return level in the omitted tide gauge location. This procedure was repeated over all tide gauges apart from Kemi and Hamina, as our models are not suitable for spatial extrapolation. We then calculated bias and mean absolute error (MAE) for the posterior median and conditional rank probability score (CRPS; Hersbach, 2000) for the full posterior distribution against the observed 50-year return level. The resulting statistics are shown in Table 2, when averaged over the ten tide gauges. As expected, the spatial models fitted without the target tide gauge (Spline$_{loo}$ and GP$_{loo}$) have worse statistics than the "full" models. In particular, GP$_{loo}$ has the largest errors in all cases. However, absolute differences in the error statistics to the model estimates based on full data set are not large, which suggests that both the Spline and GP models are able to provide useful posterior predictions in ungauged locations."

We will also add two new panels to Fig. 2, which show the spatial distribution of 50-year return level estimates along the Finnish coast and expand the description of this figure as follows:

[Figure]

**Figure 2.** Illustration of the (top) spline and (bottom) Gaussian process fits for the (left) location and (right) scale parameter with respect to the distance from Kemi tide gauge. Darker (light) shading denotes the interquartile (95 %) range of the parameter estimates. The figure also shows box-plots of the corresponding parameters at the tide gauge locations. The bottom row shows similar plots for the 50-year return level of annual maximum sea levels for both models. The box covers the interquartile range (IQR) and the median value is highlighted by the horizontal line. The length of the whiskers is one and half times the IQR.

"The bottom row in Fig. 2 illustrates how the spatial 50-year return level estimates look like for both models. The shape parameter ξ has been sampled from the joint posterior distribution of the tide gauge specific parameter values when drawing these plots. It is seen that the spatial return level estimates vary smoothly along the coast and match relatively closely with the tide gauge specific estimates of these models. In the following, we concentrate on the tide gauge specific estimates, as it allows us to compare the results of all four models."

2. *Another motivation for using a spatial model is the reduction in estimation uncertainty. I would suggest the authors quantify and discuss this reduction in more detail. By which factor is the uncertainty reduced? Figures 6 and 7 already provide a visual indication, but I think the discussion should be more quantitative.*

*Perhaps a figure or a table showing posterior standard deviations for the 50-year return levels is all that is needed.*

We will show in a new table (Table 3) the standard deviation of the predicted 50-year return level for each model and its ratio to the Separate model for the three hierarchical models with the following discussion added to the manuscript:

| Model | Kemi | Oulu | Raahe | Pietarsaari | Vaasa | Kaskinen | Pori | Rauma | Turku | Hanko | Helsinki | Hamina |
|---|---|---|---|---|---|---|---|---|---|---|---|---|
| Separate | 85 | 75 | 66 | 59 | 67 | 91 | 86 | 85 | 67 | 67 | 92 | 85 |
| Common | 61 (72) | 62 (82) | 56 (84) | 55 (93) | 50 (75) | 49 (55) | 49 (57) | 51 (60) | 48 (72) | 43 (65) | 48 (52) | 62 (74) |
| Spline | 63 (74) | 50 (66) | 54 (82) | 48 (82) | 42 (63) | 41 (45) | 40 (47) | 38 (44) | 40 (60) | 39 (59) | 47 (51) | 59 (70) |
| GP | 61 (71) | 53 (70) | 56 (84) | 50 (85) | 45 (68) | 42 (47) | 42 (49) | 42 (49) | 42 (63) | 41 (61) | 48 (52) | 58 (69) |

**Table 3.** Standard deviation of the predictive distribution for 50-year return level (in mm), shown separately for each tide gauge and model. The percentage fraction of standard deviation with respect to Separate model is given for the three hierarchical models in the brackets.

"To further illustrate how much the hierarchical modeling approach reduces the prediction uncertainty, Table 3 shows the standard deviation of the predicted 50-year return levels in the tide gauge locations and its ratio (in percentage) with respect to the Separate model for the hierarchical models. The spread is reduced in all cases and in some locations for Spline and GP is less than 50 percent of that for the Separate model. There are no major differences between the hierarchical models, although the reduction in the predictive uncertainty tends to be slightly smaller for the Common model than for the two spatial models. This supports the conclusion that our hierarchical models are able to reduce uncertainty in the posterior predictions."

3. *I think that authors should perform an analysis of sensitivity to prior choices, especially for the parameters defining the spline and GP models. It is well known that the GP parameters (standard deviation and length scale) are challenging to estimate. Also, please explain how and why these priors were chosen.*

Following the reviewer's suggestion, we performed numerous sensitivity analyses on prior choices for all hierarchical models and checked how a narrower/wider prior distribution for the GEV distribution parameters and their hyper parameters affected the posterior distribution estimates. An order of magnitude change was made to the prior standard deviations for most parameters. For the shape parameter and Gaussian process kernel parameters, the values were halved/doubled. Setting narrower priors for the shape parameter, Spline random walk parameters and Gaussian process kernel parameters affected the posterior distributions of the GEV parameters most visibly. Increasing the width of the prior distributions, however, did not affect the results noticeably, which supports the sensibility of our original prior choices.

The priors for the model parameters are mostly conventional choices, e.g. normal and log-normal distributions, chosen in such a way that they do not affect too much on the posterior estimates, as discussed above. For some of the parameters, particularly for those defining the spatial dependency, like the correlation range in Gaussian process

prior, a somewhat more informative prior has to be chosen due to identifiability problems, which is typical in this situation as the reviewer points out. We point out that even the choice of correlation function is a part of the prior specification and should be subjected to model criticism. We have performed cross-validation sensitivity studies, and one reason for testing two different types of spatial dependency (GP and spline) is to study the robustness of the result on the choice of the prior model. A similar discussion about the sensitivity tests and prior choices will be included in the supplementary material.

4. *please show the posterior estimates (with uncertainty estimates) for all the scalar parameters (and hyperparameters) of the model, either as a plot or a table.*

We will include in the supplementary material three tables, which show the summary statistics of the posterior estimates of all scalar and hyper parameters for the hierarchical models. We decided to put the tables to the supplementary material to avoid expanding the manuscript too much.

*Specific comments:*

*Extraction of annual maxima. Was the tidal component removed prior to extracting the annual maxima from the tide gauge records?*

The effect of tide on sea level is ~10 cm at most on the Finnish coast. Therefore, the tidal component was not removed from data before the analysis.

*Equation 7. The Greek letters used to denote the GP standard deviation and length scale are different between the article and the Supplementary Information.*

Thank you for noticing this. We will change the notation in the supplementary material to match the notation in the manuscript.

*It is unclear to me what the authors mean by 'empirical estimates'. The estimates from the Bayesian hierarchical models are conditional on the observations, so they are 'empirical' too, aren't they?*

We agree that the nomenclature used in the manuscript was misleading. We will change "empirical" to "observed", when discussing return levels estimated directly from the observations.

*Please add either posterior SDs or credible intervals to Table 2.*

We will add 95% credible intervals to this table (Table 4 in the updated manuscript).

*Discussion: Line ~335. While I agree that it should be emphasized that to quantify flood risk one should include mean sea level changes, I do not think that excluding mean sea level influences is a limitation of your study, rather it is a choice to focus on the storm surge component of sea level. The actual limitation is to assume stationarity, but this is discussed in the next paragraph.*

We agree with the reviewer and will slightly change the wording in the corresponding paragraph to underline the fact that removal of mean sea level was a choice rather than a limit of this study.

*Discussion. Another limitation that is not mentioned is that the Bayesian hierarchical models used in this study assume conditional independence in the likelihood. In other words, they assume that, after accounting for dependence in the marginal GEV parameter, the annual maxima are independent across stations. However, this assumption is unlikely to hold because the stations are geographically close and thus they are going to be affected by the same extreme events, which means that the time series of annual maxima are going to be correlated between stations (what is called 'residual dependence'). Ignoring residual dependence means that your uncertainty estimates are narrower than they should be (probably only slightly), but other than that it should not significantly affect the estimates presented in the paper. This limitation should be discussed. Calafat and Marcos (2020) provide a way for addressing residual dependence, but I recognize that this is beyond the scope of this paper.*

This is true, and we will add the following text to the Discussion section to point out this limitation:

"Another limitation for our hierarchical models is that they only account for dependence in the marginal GEV parameters and do not take additional residual dependence (dependence in annual maxima between different tide gauges) into account. Exclusion of residual dependence implies that our uncertainty estimates are likely slightly too narrow. One way to address this shortcoming is provided in Calafat and Marcos (2020), who use a max-stable process to capture the residual dependence. Their approach is, however, outside the scope of this paper."

**References**

Hersbach, H. (2000). Decomposition of the Continuous Ranked Probability Score for Ensemble Prediction Systems, *Weather and Forecasting*, *15*(5), 559-570.

---

## Author Comment (AC2)

**Reply to Reviewer 2**

We thank the reviewer for his/her detailed comments on our manuscript. In the following, the original reviewer comments are given in italics and our point-by-point responses to the reviewer's comments in roman font with planned changes to the text put into quotation marks.

*The future "climate" of water levels is one of the core problems for low-lying areas. The manuscript addresses this problem by means of advanced statistical modeling of parameters of extreme value distributions for future water levels and a sort of ensemble projection of extreme water levels and their return periods.*

*The analysis is theoretically sound, relies on high-quality data sets, has been performed professionally and leads to an interesting set of results. The presentation is clear and well structured, uses correct English and brings enough details for understanding the material.*

*General comments:*

*I am thus generally happy to recommend the manuscript for publication.*

*Before sending to print, however, I recommend to expand the presentation a little bit to cover some aspects that may mislead inexperienced readers and to make a few adjustments that would make the interpretation more exact and the message clearer. The recommended changes and additions only address single wording features and interpretation aspects (most of which are technically acceptable as presented in the manuscript) and do not involve any large changes to the presentation.*

*A potential trap for some readers may be the interpretation of the limited set of arguments of the Weibull distribution. Even though the authors mention on lines 307–308 that the shown values [of the upper threshold for the argument of the reverse 3D Weibull distribution] should not be interpreted as actual limits for the sea level, I would recommend commenting the related aspects in more detail to make the situation clear. There are two aspects worth of mentioning.*

*Firstly, the limited region of validity of the 3-parameter Weibull distribution could be interpreted differently. On the one hand, there is a temptation to think that this distribution provides the final truth about some properties of the described processes. On the other hand, the existence of this kind of threshold is not really physical and could be interpreted as showing that the entire GEV approach loses its validity near and behind this threshold.*

*Secondly, the set of block maxima may contain elements of different water level "populations" of the Baltic Sea. The reason is the well-known property of the Baltic Sea:*

*its water volume may increase or decrease considerably for several weeks by water exchange through the Danish straits. The "population" of the background water level of the Baltic Sea roughly follows a Gaussian distribution whereas the local storm-driven surges roughly follow an exponential distribution [Soomere, T., Eelsalu, M., Kurkin, A., Rybin, A., 2015. Separation of the Baltic Sea water level into daily and multi-weekly components. Continental Shelf Research, 103, 23–32, doi: 10.1016/j.csr.2015.04.018]. It may thus easily happen that the block (annual) maxima do not necessarily come from the same distribution. In this case the GEV distribution is just a passable approximation of the distribution of the block maxima and nothing more. It may easily be that the large scatter of the threshold of question is a reflection of this feature.*

*In this sense it is better to remove the conjecture "From theoretical perspective, this suggests that there might be an upper limit that the sea level extremes can reach along the Finnish coast" on lines 369–370 from the manuscript and also to modify the sentence "This also suggests that the hierarchical models can be used to estimate theoretical upper limits of the extremes of short-term sea level variations along the Finnish coast" to make sure that the unexperienced readers are not mislead.*

We had similar concerns about providing quantitative estimates for the theoretical upper limit in the manuscript. It indeed might give a false impression for the reader that this limit would be an actual physical upper boundary for annual sea level maxima in the study region, which it certainly is not. Thus, we have removed the comment about the theoretical upper limit from the Conclusions section and rephrased the similar sentence in the abstract to make it more neutral.

We will also comment on the alternative approach to using GEV distribution as a model for sea level extremes in the Discussion section as follows:

"We have used GEV distribution to model the overall annual sea level maxima (after removing long-term trends). However, different processes contribute to variations in sea level in the Baltic Sea and cause it to fluctuate at different temporal scales. Therefore, alternative modeling approaches could be considered to model separately the different sea level fluctuations and their contributions. For example, Soomere et al. (2015) have used an approach in which separate statistical models were fitted to weekly-scale and local storm-surge driven sea level fluctuations".

*Specific comments:*

*The Abstract seems too long, e.g., the sentence on lines 3–5 could be removed without any loss to the message and the material on lines 11–13 could be made more compact and smooth.*

After considering this comment, we slightly modified lines 11–13, but decided to keep lines 3–5 as they were. We felt that otherwise the abstract would have lost part of the motivation for the hierarchical modeling approach.

*Line 21: probably "associated WITH" or similar.*

Corrected.

*Lines 23–24: even though the increase in the mean sea level has exceeded the global average during the past 50 years in the Baltic Sea in many locations, there are opposite examples, e.g., the sea level on the Latvian shores [Männikus, R., Soomere, T., Viška, M. 2020. Variations in the mean, seasonal and extreme water level on the Latvian coast, the eastern Baltic Sea, during 1961–2018. Estuarine Coastal and Shelf Science, 245, Art. No. 106827, https://doi.org/10.1016/j.ecss.2020.106827]. This feature very shortly reflected in (Weisse et al., 2021) and may easily be overlooked. Also, it seems to have local character.*

We will include a remark about this in the same location:

"... with some local exceptions from this trend (e.g., Männikus et al., 2020)."

*Line 33: it is recommended to insert a reference to the analysis of meteotsunamis in the study area even though such a reference appears later.*

Done.

*Lines 38–39: while piling up water in the ends of the Bay of Bothnia and Gulf of Finland for sure is one of the main reasons for very high water level in these locations, the role of piling and emptying the entire subbasin is probably minor there compared to harbor-type oscillations. See, for, example [Jonsson, B., Döös, K., Nycander, J., Lundberg, P. 2008. Standing waves in the Gulf of Finland and their relationship to the basin-wide Baltic seiches. Journal of Geophysical Research-Oceans, 113 (C3), C03004, doi: 10.1029/2006JC003862]. Still, this effect seems to be a decisive one in some other basins, such as the Gulf of Riga [Männikus, R., Soomere, T., Kudryavtseva, N. 2019. Identification of mechanisms that drive water level extremes from in situ measurements in the Gulf of Riga during 1961–2017. Continental Shelf Research, 182, 22–36, doi: 10.1016/j.csr.2019.05.014.].*

We will slightly modify this sentence following the reviewer's comment:

"The largest sea level variations on the Finnish coastline take place in the ends of the Bay of Bothnia and Gulf of Finland due to the piling up effect and standing wave oscillations within the bay (Jönsson et al., 2008)."

*Line 61: "However, they did not consider spatial dependencies explicitly in their analysis" is ambiguous and is better to be removed.*

Done.

*Line 80: consider replacing "extends" by "applies".*

Done.

*Line 116: "which should reduce the correlation between the annual maximum values." is of course correct but this operation most likely almost totally removes this correlation.*

We will change "*which should reduce the correlation*" to "*which removes the correlation*" to better underline the removal of correlation between the annual maxima by this operation.

*Line 139: What is the meaning of the plus sign at the end of square brackets?*

The plus sign denotes that the function is defined only if the term within the square brackets is larger than zero.

*Line 141: consider replacing "y has bounded upper tail" (that is mathematically nonsense for an argument) by perhaps a longer explanation that the GEV distribution function is only defined until a specific value of y which is often associated with the theoretical maximum or minimum value of the process under consideration.*

We prefer to keep the explanation short but will replace "y has bounded upper tail" with "y has an upper limit".

*Line 173 and in several locations below: the simple use of "Common" (or similar) makes reading fairly complicated. Consider using "The COMM version/approach/ model" etc., e.g., as on line 246.*

Thank you for this comment. We discussed changing the names of the methods, but decided to keep the original naming convention for both the Common and Separate model.

*Line 199: Do you have a specific reason for using norm when evaluating the expression in square brackets?*

We will change the norm to parentheses. There is not any particular reason for using a norm in this case.

*Line 234: consider treating "elpdLOO" as a variable, e.g., $elpd_{LOO}$ unless you have reasons for using the text mode. Anyway, unify the use of $P_{LOO}$ in Table 1 and as text on line 243.*

We have loosely followed the notation style used in Vehtari et al. (2017) (upright for elpd and italics for p) but will unify the use of $p_{loo}$. As a minor modification, we will change the subscripts and both symbols to lowercase.

*Line 248: probably "location and scale PARAMETERS" are meant.*

Corrected.

*Line 253: it is not recommended to start the sentence from a symbol or expression.*

Corrected.

*Line 267: consider expanding the expression "Weibull-type distribution" towards explanation that the GEV approach uses so-called reversed 3-parameter Weibull distribution (and not, e.g., the 2-parameter Weibull distribution that is common in the description of wind speed, wave heights, etc.). Just to make clear the scene for the reader.*

We have changed "Weibull-type" to "3-parameter Weibull" in this sentence to avoid any confusion with the 2-parameter Weibull distribution.

*Line 269–270: "they used ocean model output instead of observations in their analysis" is only partially true. They also used measured data from five locations and noted strong variations in the shape parameter depending on both the particular location and the method for evaluation of the parameters of the GEV distribution.*

This is true and will modify this paragraph as follows:

"...A slightly contrasting result was obtained by Soomere et al. (2018), who estimated shape parameter values close to zero from an ocean model output along the Estonian coast. Furthermore, in their additional analysis based on tide gauge observations the shape parameter varied strongly depending on the location and method used to estimate the GEV distribution parameters..."

*Line 276: "separate fits": see comment to line 173.*

Please see our reply above.

*Section 6 Conclusions: it is recommended to remove the short names of scenarios from the text in order to make the section readable on its own.*

Done.

*References:*

*Coles 2001/2004 is missing from the list*

Thank you for noticing this! The reference was lost due to an unknown reason, when we compiled the pdf from the latex template.

---

## Author Response (AR2)

**Reply to Reviewer 1**

We thank the reviewer for his/her comments on our manuscript, which have further helped us to improve it. In the following, the original reviewer comments are given in italics and our point-by-point responses to the reviewer's comments in roman font with changes to the text put into quotation marks.

*I have now read through the revised version of the manuscript. I want to thank the authors for addressing all my questions. I am satisfied with the paper as it stands and thus recommend publishing it. I have only two minor comments, which are outlined below.*

*1. In Table2, I would suggest the authors add some kind of relative metric to convey a sense of the magnitude of the errors relative to the 50-year return level (e.g., Bias/return_level).*

We have added relative bias defined as the reviewer suggests to Table 2.

*2. I am still a bit confused about how the authors calculate what they call "observed return level". Could you please clarify?*

We have further clarified how the observed return level is estimated in the text as follows (lines 254–256):

"The observed 50-year return level has been estimated by first calculating the exceedance probability for the observed annual maxima based on Weibull plotting positions and then interpolating the estimated probabilities between the observed values."

In addition, we also note that Weibull plotting position was not used in the previous manuscript version and the estimation was based on the default settings for *ppoints* function in R. This change has slightly modified the numerical values in Table 2 and Figs. 3, 4 and 7 and led to small changes in the text. However, the results and conclusion are essentially the same.